# Smart Spare Parts (SSP) in the Context of Industry 4.0: A Systematic Review

**DOI:** 10.3390/s24165437

**Published:** 2024-08-22

**Authors:** G. Morales Pavez, Orlando Durán

**Affiliations:** 1Digital Fabrication Laboratory, School of Mechanical Engineering, Pontificia Universidad Católica de Valparaíso, Av. Los Carrera, Quilpué 01567, Chile; 2Maintenance and Reliability Laboratory, School of Mechanical Engineering, Pontificia Universidad Católica de Valparaíso, Av. Los Carrera, Quilpué 01567, Chile

**Keywords:** smart part, additive manufacturing, embedded sensor, Industry 4.0, systematic literature review

## Abstract

The implementation of Industry 4.0 has integrated manufacturing, electronics, and engineering materials, leading to the creation of smart parts (SPs) that provide information on production system conditions. However, SP development faces challenges due to limitations in manufacturing processes and integrating electronic components. This systematic review synthesizes scientific articles on SP fabrication using additive manufacturing (AM), identifying the advantages and disadvantages of AM techniques in SP production and distinguishing between SPs and smart spare parts (SSPs). The methodology involves establishing a reference framework, formulating SP-related questions, and applying inclusion criteria and keywords, initially resulting in 1603 articles. After applying exclusion criteria, 70 articles remained. The results show that while SP development is advancing, widespread application of AM-manufactured SP is recent. SPs can anticipate production system failures, minimize design artifacts, and reduce manufacturing costs. Furthermore, the review highlights that SSPs, a subcategory of SPs, primarily differs by replacing conventional critical parts in the industry, offering enhanced functionality and reliability in industrial applications. The study concludes that continued research and development in this field is essential for further advancements and broader adoption of these technologies.

## 1. Introduction

Technological advances in areas such as additive manufacturing, electronics, and engineering materials allow for the implementation of new strategies in the production of smart parts (SPs). An SP is characterized by its ability to perform a series of functions through the integration of sensors and communication systems. This enables it to provide data for wirelessly monitoring working conditions and anticipating catastrophic failures in production processes [1,2]. The design of a smart part (SP) must involve internal cavities that allow the integration of sensors and other electronic devices. However, the challenges posed by conventional manufacturing processes significantly limit the production of SPs, making it difficult to construct complex internal geometries and introduce sensors [1,3,4].

Similarly, additive manufacturing addresses the complexities of conventional processes due to the flexibility it offers during production. In other words, it is possible to pause the printing process to insert a sensor or an electronic component and then resume material deposition over the inserted element. This technique is known as “Stop and Go”, which is used to incorporate various elements [1]. Despite its great potential, additive manufacturing presents challenges such as high-temperature processing, material expansion, and the effectiveness of the sensor or electronic device within the SP, among others [1,2,3,4,5,6].

Nevertheless, various types of sensors can be incorporated into the cavities of SPs. For example, inductive proximity sensors have shown good performance when packaged in ceramic parts, as they have demonstrated the ability to function satisfactorily at high temperatures [7]. Additionally, piezoelectric sensors have been characterized inside parts manufactured using electron beam melting (EBM). This characterization has enabled the measurement of compressive force capacity by detecting voltages, thus monitoring the performance of the component [1].

The characterization has enabled the measurement of compressive force capacity by detecting voltages, thus monitoring the performance of the component. Another topic of interest is the incorporation of communication systems within SPs that transmit information wirelessly, avoiding the integration of electronic circuits and wiring that could reduce the performance of these electronic components. For instance, SPs have been developed by incorporating radio frequency identification (RFID) antennas to transmit information wirelessly, integrating it into a single process. The results have shown that the electrical circuits are limited by high temperatures and the recording and transmission of vibrations at higher frequencies [8].

For an SP to be functional, it is essential to select a material that meets the environmental demands. For instance, 16MnCr5-grade steel is known for its high wear resistance and is commonly used in the manufacturing of shafts, gears, and sprockets [8]. Additionally, titanium alloys are known for their excellent corrosion resistance and high specific strength compared to other materials such as steels [1]. The limitations that materials impose on the fabrication of SPs are associated with the sintering temperature, which can exceed the melting point of the sensor [7].

By integrating sensors within parts, it is possible to measure parameters such as temperature, pressure, force, and flow, thereby estimating the operating conditions of the component. Indeed, real-time monitoring of the part’s condition allows for the anticipation of component replacement, preventing outcomes that could be detrimental to production processes [9].

While SPs have been developed using additive manufacturing, the question to address is whether this concept can be applied to parts that are critical in production processes, integrating the SSP concept. Based on the above, and more generally, what are the main challenges in implementing SPs in Industry 4.0? One area of contribution for SPs is in productive industries such as energy, automotive, agriculture, and transportation, to name a few [7,9]. In this way, SPs would enable real-time monitoring of productive industries and help prevent costly damages [8,9].

Regarding the general objective of the systematic review, it focuses on conducting an exhaustive synthesis of the available scientific articles on the fabrication of SPs using AM. The purpose is to critically evaluate and analyze the existing evidence to answer the research questions.

The structure of the document is as follows: Section 2 outlines the methodology applied for the systematic review, emphasizing the search strategy. Section 3 presents the results, addressing the research questions. In Section 4, discussions are conducted on the findings and the advancements achieved with the concept of the smart spare part.

## 2. Methodology

Figure 1 shows the area where the knowledge domains related to SPs converge and underpin the systematic review. This convergence allows for the creation of a reference framework from various disciplines for the systematic review.

To limit the search, it is important to identify a series of research-related questions. This can be broken down into a general question (GQ) and research questions (RQs). Specifically:GQ: What are the main challenges in implementing smart parts (SPs) in Industry 4.0?RQ1: What are the predominant factors that make additive manufacturing (AM) an ideal method for the construction of SPs?RQ2: What are the limitations of AM in the construction of SPs?RQ3: Do sensors and electronic components reduce their performance when incorporated into a part manufactured with AM?RQ4: What types of materials are most suitable for the fabrication of SPs using AM?RQ5: Is there any substantial difference between SPs and smart spare parts (SSPs)?RQ6: What economic impacts can SPs generate in production processes?

By establishing the reference framework associated with the areas that impact the development of SPs and limiting the search through the questions, the first phase of planning and development was generated. The second phase was the search for sources and scientific documents. It was important to include a wide variety of articles to reduce bias and validate the systematic search. Based on the questions, a set of documents were detected using different databases, a specific language, and a specific time period. Additionally, not only were publications reviewed, but also websites, conferences, and reviews. Table 1 lists the inclusion criteria.

Additionally, the second phase includes the search for keywords related to the areas of interest and the research questions. This is executed using Boolean operators and a series of synonyms for additive manufacturing, smart part, engineering materials, and integrated sensor. Table 2 shows the different synonyms of the main keywords.

Keywords were merged considering the most relevant specific areas, which resulted in a search outcome of 1603 scientific articles. Table 3 shows the different combinations that allowed us to achieve this result.

In the identification phase, 398 duplicate articles were removed. Additionally, 90 articles without a DOI were discarded. In the second phase, the screening stage, 851 articles were eliminated based on their title and abstract. These articles did not contain the keywords for additive manufacturing and smart parts, reducing the total to 267 articles. In the “reports not retrieved” stage, 78 articles were removed due to access and payment restrictions, which was a critical phase as some of these articles were of interest. Finally, in the “reports assessed for eligibility” stage, 119 articles related to topics such as reviews, 4D printing, medicine, and other irrelevant subjects, as well as those with different objectives and applications, were eliminated. In the end, 70 articles were included in the study. The summary and the flow diagram are shown in Figure 2.

Based on the scientific articles obtained, the structure of the categories of analysis (CAs) that responded to the research questions was as follows:SP manufacturing with AM: this section discusses articles that have manufactured SPs and the considerations and limitations of manufacturing SPs with AM.Incorporation of sensors and electronic components in AM for SP manufacturing: this section describes the procedures and steps to introduce a sensor or electronic component into an AM manufactured part.Scenario of SP manufacturing by AM and the impact on industrial processes: the technical and economic impacts of including smart parts in production and industrial processes are discussed.Differences between an SP and an intelligent spare part (SSP): the differences between an SP and an SSP are explained.

Each of the categories of analysis responded to the research questions. Table 4 below associates the research questions with the categories of analysis.

## 3. Results

The first category of analysis (CA1) addressed is SP manufacturing with AM. In section CA1, additive manufacturing is defined taking into consideration design and manufacturing aspects, as well as physical, chemical, and mechanical properties. From this, the advantages and disadvantages of additive manufacturing in SP fabrication can be deduced.

Then, the second category CA2 is discussed, which describes the sequence and procedure for the introduction of elements into the SP. Third, the technical and economic impacts of including SPs in production processes are discussed (CA3). Finally, the differences between an SP and an SSP are stated (CA4).

### 3.1. CA1: Manufacturing SP with AM

Before manufacturing a part with additive manufacturing, it is important to consider that a solid must be generated from digital modeling and manufacturing computational tools. Thus, it is possible to generate the code that allows 3D printers to manufacture the part autonomously [10].

Additive manufacturing is a manufacturing process that involves the layer-by-layer deposition of material using various techniques, including but not limited to, methods that employ a nozzle or a reloading arm. The motions are typically performed in the x–y plane by a gantry containing a nozzle or laser. Once one of the layers is generated on the build platform, a movement driven by a servomotor allows the displacement in the *z*-axis of the platform, thus generating the successive layers that will form the 3D part. Other methods, such as DLP and binder jetting, also contribute to the diverse range of additive manufacturing techniques [10].

There are a total of seven types of printing processes or AM according to ISO/ASTM 52900 [11]. These are Photopolymerization (SLA), Fused Material Deposition (FDM, FFF), Powder Bed Melting (DMLS, EBM, SLS, LPBF), binder jetting (BJ), Material Jetting (MJ), Sheet Laminating (LOM), Direct Energy Deposition (DED), and a hybrid process in which additive manufacturing is mixed with subtractive manufacturing. Each of these is differentiated by the types of materials that can be used and the energy used to change the material state [10,12,13].

It should be noted that AM, like other conventional manufacturing processes, has certain limitations associated with phenomena intrinsic to the process. The following are parameters that were analyzed in the systematic review and that impose restrictions when manufacturing SPs:Surface finish.Material expansion and shrinkage.Mechanical performance.Influence of the printing material on the sensor performance.Influence of the working and sintering temperatures on sensor performance.Materials with electrical and electronic properties.

In terms of surface quality, tolerance, electrical properties, mechanical performance of the part, these depend on different factors such as printing speed, bed and extruder temperature, fill density, layer height, printing material, among others [14,15,16]. Likewise, the combination of each of the parameters mentioned above allows the manufacture of components with a lower density but sacrifices some properties such as mechanical properties [17]. The above is reiterated for other properties such as thermal, electrical and electronic properties, since the non-continuity of the material and the gap between layers reduces the properties [16,18]. While electrical and electronic properties are highly material-dependent, physical and technological advances have made it possible to formulate materials with characteristics [19].

#### 3.1.1. Surface Finish

Surface finish is a characteristic intrinsic to the manufacturing process and defines in quantitative terms the roughness of a surface. This parameter influences mechanical, electrical, and electronic properties, but by modifying the manufacturing variables, it can be decreased or increased, depending on the purpose of the part [13].

For example, in the manufacture of inkjet (BJ) printed boards, a phenomenon called contact angle hysteresis is generated, which is the resistance to contact line retraction. This hysteresis is mainly due to surface roughness and reduces the conductivity performance of the conductors [3].

It also influences the performance of sensors and the manufacture of electronic components in SPs, as roughness slightly reduces the resonance frequency in RFID antennas [8].

#### 3.1.2. Material Expansion and Shrinkage

All materials are sensitive to temperature changes, which can cause part distortion, cracking, dimensional losses, and metallurgical changes in the manufacturing processes. In additive manufacturing of smart parts, one notable occurrence is the expansion of polymeric and metal materials with increased temperature and their shrinkage when the temperature decreases [15,20].

To avoid the problems caused by dimensional changes generated in materials by temperature, it is important to consider the expansion coefficients of the materials to be used. For example, there are materials with large differences in their coefficients of thermal expansion such as platinum (metal) and zirconium (ceramic). To prevent the generation of cracks in both materials, it is important to consider the expansion of platinum and zirconium in the design [7].

#### 3.1.3. Mechanical Performance

The mechanical properties of 3D-printed parts depend on several parameters such as layer height, infill percentage, printing direction, densification, and temperature, to name a few [14,21,22]. Although it depends on the printing parameters, many times, they can undergo heat treatment where the mechanical properties of the parts can be increased [8,23]. It should be noted that heat treatments are processes that work at high temperatures and can cause damage to the sensor such as cracking and even a significant loss of transmission [24].

#### 3.1.4. Influence of the Printing Material on the Sensor Performance

Although there are materials that do not affect the performance of sensors, we focus on metals, polymers, ceramics, and composite materials that can influence the performance of sensors and electronic components [25,26]. Thus, iron–carbon alloys reduce the reading distance of inductive sensors by influencing the magnetic field [8]. The study showed that distance influences RFID performance, with greater distances resulting in lower emitted frequencies. Simulations and experiments have confirmed that the optimization of antenna parameters is crucial for maintaining effective RFID performance [8]. Additionally, the dielectric properties of polymers negatively impact radio frequency (RF) signals and RFID performance, as they attenuate the signals [25].

In another approach, in ultrasonic additive manufacturing (UAM) processes, the use of electrically conductive cooling materials can generate an impedance mismatch in the sensor. This negatively affects the accuracy of voltage measurements and the linearity of the sensor [27].

On the other hand, composite materials are generated from the union of materials from different families (metal, ceramic, polymer). In additive manufacturing (AM), fiber optic sensors can be fabricated and applied to measure structural health. Although the measurement results have been positive, it is crucial to consider the thermal expansion coefficients of the different composites to avoid damaging the fiber [26].

#### 3.1.5. Influence of Working and Sintering Temperature on Sensor Performance

DMLS, LPBF, and SLS additive manufacturing operate in temperature ranges that can cause damage to the sensor [28]. Not only can the manufacturing process generate complications related to temperatures, but also postmanufacturing processes such as sintering [7]. Sintering is a manufacturing post-process that uses high temperatures and pressure to compact and shape a part. It also improves density and eliminates moisture and polymeric binders [29].

Sintering temperatures and thermal expansion coefficients must be considered to avoid damage to the sensors. After adjusting the dimensions and performing several tests, positive results can be obtained [7,28]. One study demonstrated that sintering did not influence the sensor performance. Additionally, it was shown that good temperature control did not reduce the performance of force sensors. Even parameters such as hysteresis and linearity were not affected [7]. The inductance varied as a function of the distance at the various stages of manufacture, confirming that the sensing element performed consistently through different packaging stages, including green and sintered packaging.

#### 3.1.6. Materials with Electrical and Electronic Properties

Additive manufacturing can revolutionize the manufacture of passive and active electronic components, as well as interconnects and printed traces. This manufacturing process has the potential to significantly transform the electronics industry, especially in areas such as wireless communication, flexible electronics, efficient batteries and solid-state display technology [30].

Materials with conductive properties being used in 3D printing include silver nanoparticles, nano silver inks and isotropic conductive adhesives (ICAs). In addition, semiconducting materials such as carbon nanotubes (CNTs) and dielectric materials are being used [30].

These materials are opening up new possibilities in the manufacture of electronic components through additive manufacturing technologies, enabling innovations in the design and production of more efficient and sophisticated devices [30].

Also, polymeric materials such as thermoplastic polyurethane (TPU), polydimethyl-siloxane (PDMS) and light-curing polymers are being used in flexible electronics. These materials enable the creation of electronic devices that are more versatile and adaptable to various applications, opening up new possibilities in the design and functionality of flexible technology. Additionally, a collaborative study by UC Berkeley and NCTU demonstrated the use of liquid metals to create passive components like resistors, inductors, and capacitors, highlighting their benefits for flexible circuits and SP applications [30].

In addition, some polymeric materials, such as PEDOT on Mylar substrates, can be used for electromagnetic interference shielding, which is vital to protect electronic devices and sensors that may be affected by electromagnetic fields [31].

Although a limitation of 3D printing is its speed, some material injection molding additive manufacturing processes have fabricated piezoelectric–pneumatic materials (PPMJ) at speeds up to 500 mm/s. These processes achieve surface roughness values on the order of 2.99 μm and resistivity values as low as 0.41 Ω-cm, outperforming extrusion-based manufacturing processes [32].

As for the metallic materials used in 3D printing, platinum (Pt) is used for temperature sensing due to its resistivity characteristics. Importantly, this resistivity depends on the sintering temperature, reaching a minimum value at 1000 °C. In addition, platinum is notable for its stability at high temperatures, which makes it ideal for demanding applications where accuracy and reliability in temperature measurement are required [33].

Similarly, low-melting-point metal alloy filaments have a high conductivity [5]. Non-eutectoid alloys can be used for the manufacture of integrated circuits avoiding, for example, the stopping of the AM process to introduce such electronic components and reducing the manufacturing time [5].

Despite all the technological advances, there are still a number of challenges that need to be addressed for the application to be effective. Table 5 below summarizes the considerations and limitations of AM analyzed in the systematic review.

### 3.2. CA2: Incorporation of Sensors and Electronic Components in AM for SP Manufacturing

AM has the particularity of generating internal cavities to introduce components without the need to use attachments and manufacture complex matrices that are costly and laborious to produce. One of the techniques to introduce sensors is called “Stop and Go”, which consists in stopping the printing with the 3D printer command, then the platform is lowered to introduce the sensor. Once the sensor is incorporated, the platform resumes the initial position with the help of the operator, to resume the material deposition process and to be able to completely generate the SP. From the above sequence, the external part, which can be a sensor, is completely covered with material and can be used in conditions that can be extreme [7,34,35].

This technique cannot be applied in conventional processes because they are continuous processes and require accessories to introduce a sensor inside the part [7]. In addition to the above, the force of the flow in processes such as casting causes the sensor to lose its position, and it is often out of service [7]. However, there are difficulties in integrating sensors inside parts manufactured with AM; moreover, design and manufacturing aspects must be considered for a correct assembly. In summary, to be successful, the “stop and go” technique requires five stages as shown in the following Figure 3.

To perform the procedure correctly, it is important to describe each of the stages considering the design and manufacturing aspects. Table 6 below shows each of the aspects mentioned above.

The “Stop and Go” technique, to be effective, requires the use of various fixing methods, including mechanical and adhesive methods [39]. The use of industrial adhesives, the design of slots to protect the sensor, and the orientation of the integration are relevant for the sensor to work efficiently [38].

The “Stop and Go” method is not the only technique for inserting a sensor inside a part; there are other methods. One involves coating the fiber optic materials with a metallic layer to protect them from extreme heat during the additive manufacturing process. In this way, the optical fiber is introduced into the metal part during the selective laser melting (SLM) or direct laser melting (DML) process [26].

Other techniques for manufacturing smart components can involve multiple additive manufacturing processes. For example, the combination of laser powder bed fusion (LPBF), direct ink writing, and ultrasonic additive manufacturing (UAM) has enabled the development of sensorized parts. This integration of technologies allows the unique advantages of each process to be leveraged, resulting in more advanced and functional components [39].

The main advantage of inserting a sensor inside a part is that the sensor is not affected by environmental extremes and operating conditions, preventing a reduction in its performance. This protection allows the sensor to operate more reliably and accurately, extending its service life and improving the quality of the data collected [39].

### 3.3. CA3: Scenario of SP Manufacturing by AM and Impact on Industrial Processes

Despite the limitations of additive manufacturing (AM) in the manufacture of sensorized parts (SP), the outlook is encouraging. Studies have shown that sensors incorporated inside parts manufactured with AM, in some cases, do not reduce their capacity. In addition, the incorporation of piezoelectric sensors inside the parts allows the measurement of pressure and temperature variables under extreme conditions, facilitating their application in industrial processes such as power generation, aerospace, automotive and robotics, among others.

Based on the systematic review, the technical and economic impacts of additive manufacturing on the production industry are analyzed below.

#### 3.3.1. Technical Impact

The implementation of additive manufacturing in soft robot manufacturing allows grippers to be developed quickly and accurately, integrating sensors more efficiently compared to traditional molding techniques [40]. In addition, the in situ integration of sensors reduces additional post-processing steps, increasing efficiency and reducing accuracy errors that can be generated in other conventional processes [41]. Thanks to these advantages, the robotics industry and other related sectors can benefit significantly [40,41,42,43].

In industry, it is essential to accelerate and improve precision in the manufacture of mechanical components. Additive manufacturing has enabled the development of primary air fans (PAFs) with significant advantages over conventional processes. These advantages include the production of primary fans without surface defects, with a chemical composition and mechanical properties similar to those obtained using traditional methods and high production speed [44].

Currently, industries such as energy, agriculture, automotive industries, to name a few, are implementing structural health monitoring (SHM), condition-based maintenance, and predictive maintenance techniques [45,46,47,48,49]. In the same way, in order to obtain information from the process and apply these techniques, it is imperative to mount various measuring instruments and sensors in order to collect the data. Thus, by having the information in real time, it is possible to minimize early failures during the development phase and during operation [47,48,49,50].

Another important aspect is the measurement of parameters such as temperature, pressure and force in areas that can be complex and difficult to access [51]. Some examples are wind turbines, gas turbines, steam turbines, and nuclear power plants where, due to operating conditions, it is impossible to obtain data on the parameters [48,50]. In addition, real-time measurement and overriding inspection protocols by sensing critical components would reduce the time to access system information [50].

Additive manufacturing can have a significant impact on the manufacture of customized components in a decentralized manner. It enables the development of electrochemical sensors used in public health to detect hazardous and highly toxic compounds, such as metal ions, inorganic contaminants, and small organic molecules. This offers a fast and efficient solution to address health and safety issues, highlighting the versatility and innovative capability of additive manufacturing in the creation of specialized, high-performance devices [52].

Finally, one of the areas of greatest impact of additive manufacturing is in electronics and electrical manufacturing. Significant advances in filaments and materials used in additive manufacturing have enabled the fabrication of functional parts with specific electrical properties. These advances change the way electronic components are designed and produced, offering new possibilities for the creation of more efficient, customized devices and reducing material waste [18,53,54,55,56,57].

#### 3.3.2. Economic Impacts

Additive manufacturing allows the fabrication of parts with electrical and electronic properties that can also interact with and measure certain phenomena. For example, techniques such as solvent evaporation-assisted manufacturing have enabled the one-step fabrication of piezoelectric devices with nanocomposites. This simplifies the process by eliminating additional steps, such as electrical polarization, thus improving the efficiency and functionality of the devices and reducing processing costs [58].

There are combinations of techniques for the fabrication of piezoelectric sensors that can be used in additive manufacturing for the creation of smart parts. One of these combinations involves the use of additive manufacturing by extrusion along with electrospinning to create piezoelectric sensors in a single step. This integration eliminates the complications and time associated with multiple assembly methods, as well as reduces manufacturing costs [59].

In the manufacturing of thermocouples, the use of additive manufacturing also contributes to reducing production costs. This is due to the elimination of the need to manufacture molds and specialized tools for this component. Additionally, by directly integrating the sensor during the additive manufacturing process, the need for additional assemblies is reduced, simplifying the supply chain. As a result, material costs, labor, and production times are reduced [60,61,62,63].

Although additive manufacturing reduces costs associated with assembly stages, mold fabrication, and tool acquisition, it comes with a high initial investment cost. Additionally, it requires specialized labor, and operations must be fully automated. Compared to other conventional processes, the initial investment in additive manufacturing is higher and specialized labor is not always required for these traditional processes [44].

Another critical factor in additive manufacturing is the high cost of materials with functional applications. For instance, alloys like Hastelloy X are priced at approximately 4000 USD/kg, Inconel 718 costs around 3000 USD/kg, and Ti6Al4V costs about 1990 USD/kg. These prices significantly exceed the costs of these materials when manufactured using other manufacturing processes [64].

Energy consumption is another critical factor to consider when evaluating costs in additive manufacturing. In fused filament fabrication (FFF) processes, energy consumption can range from 62 Wh to 304 Wh, and it is significantly higher in other processes such as powder bed fusion. In FFF processes, the bed heating strategy is particularly crucial as it has the greatest impact on energy consumption [65].

However, there are studies analyzing the impact of additive manufacturing on costs. For companies looking to manufacture functional prototypes of smart parts, open-source 3D printing can be a cost-effective option. This technology does not incur high initial investment costs and allows for ROI within five years, with simple payback in less than six months for high-cost products [66].

### 3.4. CA4: Differences between a Smart Part (SP) and a Smart Spare Part (SSP)

The scientific articles analyzed in the systematic review define a sensorized part, which allows real-time monitoring of environmental conditions, as an SP [1,67]. Furthermore, by applying the same technique, it is possible to develop intelligent manufacturing processes, enhancing efficiency and real-time control of various variables, as demonstrated by the study on hybrid interconnections for 3D-printed electronics in harsh environments, as well as condition monitoring in foundry processes. Additionally, the data obtained by the sensors can be used to simulate the process, bringing it closer to reality [68,69]. However, it is essential to discuss the definition of an intelligent part within a maintenance and condition monitoring environment, rather than in a general manner. This is because, considering the productive, economic, and safety impacts, the part must not only withstand environmental conditions but also have the capability to indicate the appropriate time for its replacement. Taking into account the aforementioned conditions and characteristics, we define an SSP.

These spare parts allow real-time monitoring of operating conditions, ensuring system integrity and performance in situations where conventional maintenance would be complicated or impossible [70]. Additionally, the mechanical component replaced by the SSP must be critical and interchangeable, as its failure in the production process would have significant economic and operational impacts. In summary, SSPs have the following characteristics:They are sensorized and interchangeable parts.They can transmit information wirelessly, reducing or eliminating wiring altogether.They are implemented under extreme conditions such as high temperatures, high pressure, irradiation, among others.They are manufactured using subtractive and additive manufacturing. The latter is attractive due to its manufacturing flexibility and low cost.They allow real-time process monitoring.They are highly critical parts, and their failure can significantly impact production processes due to their high costs.They have the ability to indicate when they need to be replaced using algorithmic and predictive techniques, in conjunction with information obtained from sensors.They reduce downtime and optimize costs by planning replacements, thereby allowing for adequate inventory management.

Based on the analysis conducted in the systematic review, it is concluded that all SSPs are SPs, but not all SPs are SSPs (Figure 4). This implies that not all smart parts are designed to be spare parts. While some SPs may be integrated into a mechanism without the additional characteristics expected of an SSP, such as self-diagnostic capabilities to determine when they need to be replaced, others do possess these features. For instance, sensors can be integrated into a gear to monitor vibrations and send real-time data to prevent failures. In contrast, a gear that not only monitors its own condition and performance but also alerts and plans when it needs to be replaced, and can be managed within the inventory to ensure one is always available, exemplifies an SSP.

In summary, the findings from the systematic review quantify the advancements achieved in various scientific articles. They demonstrate the efficiency of sensors measuring deformation, temperature, pressure, electrical resistance, and force within 3D-printed materials [71,72,73,74]. The vibration sensors exhibited good performance, similar to pressure sensors. However, improving the communication between the SP antenna and external receivers is necessary to achieve even more satisfactory results. On the other hand, inductive sensors showed lower performance due to their placement inside iron–carbon alloy parts, resulting in a reduced magnetic field perceived by the measuring instruments [8].

Finally, it is important to highlight the difference between an SP and an SSP. Although these concepts were not discussed in the scientific articles analyzed in this systematic review, it is crucial to differentiate them in order to establish new areas of contribution.

## 4. Discussions and Future Work

Additive manufacturing is a flexible process that allows for the fabrication of parts with complex geometries and the integration of elements using various techniques. One of the most applied techniques in this context is called “Stop and Go”. This allows the production of parts at a lower cost and faster manufacturing times than conventional processes because it does not require fixtures [75].

The results obtained in the systematic review show significant advancements and achievements in the manufacturing of SSP using additive manufacturing. Key findings are related to the implementation of RFID communication systems inside SPs, which allow for the complete elimination of wiring from the parts [8,76]. Analogously, research is being conducted to establish communication protocols, enabling interaction among sensors, machines, and systems in a production environment [76]. The importance of the mentioned attributes is that complex mechanisms with movements, such as gear trains, do not require the use of cables. At the same time, by wirelessly transmitting sensor information to the cloud, operations can be streamlined and optimized remotely [77].

Although most results were relevant, not all were successful, crucially highlighting the limitations of RFID communication systems. To ensure SSP functionality in hostile industrial environments, thermal treatments are essential [8]. On the other hand, most sensors and electronic components are limited by high temperatures, which can significantly reduce their performance [8].

Parallelly, the materials used in SSPs can degrade sensor performance; for instance, iron alloys may block electromagnetic signals from inductive sensors [8]. On the contrary, the use of ceramic, polymeric, and composite materials is compatible with the nature of inductive sensors. However, a limitation of polymeric and ceramic materials is that they cannot be applied in mechanisms such as gear trains [8].

Another finding of great interest is the use of polymeric and ceramic materials with electrical and electronic properties, enabling the design of integrated circuits in situ [18,53,54,55,56,57]. It is possible to eliminate the stages associated with the “Stop and Go” technique using multimaterial extrusion technologies (M-MEX) that allow for the deposition of different materials in the part. It is important to note that although it is an innovative technique, it may present issues related to SSP quality. Phenomena such as distortion, delamination, material incompatibility, and adhesion problems are present when working with multimaterials in AM.

To conduct the systematic review, PRISMA 2020 guidelines were employed, using various inclusion and exclusion criteria, resulting in the selection of a total of 70 scientific articles. Inclusion criteria included language, databases, sources, and relevance of the research to the research questions. While there could be bias in the search due to these criteria, detailing each step is crucial to achieving consistent results. Exclusion criteria involved aspects such as article duplication, irrelevant articles, and non-pertinent articles. The latter two criteria can be subjective, as researchers’ perspectives may differ, potentially introducing bias. To mitigate this, it is important that the research questions are clear, precise, and that the analyzed articles are relevant to the keywords.

An important aspect of the search lies in the indiscriminate use of the term SP without clear distinctions regarding its possible applications. Initially, the search was complex due to the use of a very specific keyword, in this case, SSP. However, through the systematic review, a new term was defined: smart critical spare part (SSP). As discussed earlier, this term differs from SP in the component’s importance under extreme conditions and the need to monitor the process in industrial production. Although they have not been widely used in industrial production so far, this opens the door to the possibility of manufacturing parts that meet these specifications using AM technology.

Based on the research questions posed, satisfactory answers were obtained through the analysis of scientific articles. However, it is important to highlight that question SQ5 requires further discussion with other specialists in the field. Therefore, it is relevant to establish a new definition of SSP based on the observations mentioned in the results.

The significant results observed in piezoelectric sensors, force sensors, among others, are relevant [71,72,73,74]. Comparatively, piezoelectric sensors performed better than inductive sensors under external loads. Additionally, no significant changes in sensor performance were observed when processes like sintering were applied, taking into account temperatures that could affect the sensor. This represents an initial approach of AM-manufactured SSPs towards high-demand industrial applications.

In summary, the analyses were focused on the main research question regarding the primary challenges in implementing SP in Industry 4.0. While there is still much to progress, it is important to outline some projections as listed below:Material limitationsCurrent constraints: Additive manufacturing is an appealing process for the production of smart parts; however, it is currently constrained by the materials available on the market. The existing range of materials is often insufficient for high-temperature, high-strength, and high-pressure applications, which are critical for many SSPs.Future direction: Advancing new materials specifically designed for additive manufacturing will broaden manufacturing possibilities and enhance the performance of SSPs in demanding environments.Manufacturing process complexityCurrent issues: The manufacturing process for SSPs using AM involves multiple steps, including the integration of sensors and electronic components. Each step increases the potential for errors and extends production times.Proposed solutions: Reducing the number of steps in SSP manufacturing can increase production rates and minimize errors. Innovations such as the development of more efficient multimaterial 3D printers and the discovery of new materials can facilitate this step reduction and improve success rates in component manufacturing.Sensor integration challengesDesign constraints: The integration of sensors into SSPs without significantly altering the design poses a major challenge. Large or bulky sensors can compromise the structural integrity and functionality of the part.Future research: Research efforts should focus on designing smaller, more integrated sensors and electronic boards. This will enable the seamless incorporation of sensing capabilities into SSPs without compromising their design.Behavior in extreme conditionsTesting limitations: While some studies have analyzed the behavior of SSPs in extreme conditions, such as high temperatures and pressures, there is a lack of comprehensive instrumentation protocols for such environments.Research needs: It is necessary to establish robust instrumentation protocols to ensure the reliable performance of SSPs under extreme conditions. This will involve extensive testing and validation of SSPs in real-world scenarios.Communication and connectivityCurrent technology: Implementing the concept of smart spare parts (SSPs) using additive manufacturing requires the use of various predictive algorithms to determine the optimal replacement time. These algorithms are essential for real-time monitoring and lifecycle management of SSPs. Current challenges include the need for platforms capable of handling and analyzing large volumes of data generated by these predictive algorithms, ensuring precise and timely maintenance decisions.Future direction: Future innovations should focus on developing robust data analysis platforms that seamlessly integrate with predictive algorithms. These platforms must efficiently process vast datasets to enable accurate predictions for maintenance and replacement schedules. Additionally, designing smaller, more integrated sensors will enhance the data collection and transmission capabilities of SSPs, thereby improving their reliability and effectiveness within Industry 4.0 frameworks.Predictive algorithms and data analysisCurrent technology: The communication of SSPs with equipment, instruments, controllers, and cloud-based systems (cloud computing) needs improvement. The integration of RFID antennas and other communication technologies is essential for the real-time monitoring and management of SSPs.Material and design innovations: This requires the design and use of special materials that can operate under specific conditions, ensuring reliable communication and data transfer. Future innovations should focus on enhancing the connectivity and integration of SSPs within Industry 4.0 frameworks.

By addressing these challenges through focused research and technological advancements, the development and implementation of smart spare parts using additive manufacturing can be significantly improved, leading to more efficient and reliable maintenance solutions in various industries.

Furthermore, it is essential to discuss the limitations of additive manufacturing in the production of SSPs based on a systematic review:Manufacturing size: Additive manufacturing processes are limited by the size of the parts that can be produced. There are critical large-dimension components that cannot be manufactured through this process.Materials: although there is a wide range of materials that can be used in additive manufacturing, it is still insufficient for more critical applications, such as those requiring high temperature, strength, and pressure resistance.Production speed: production speed is a limitation, as it is currently only suitable for the manufacturing of low-turnover parts.Integration of sensors and electronic components: the integration of sensors and electronic components within the parts may require multiple steps, increasing the likelihood of failure and thus extending manufacturing times.Quality control in the “Stop and Go” technique: when applying the “Stop and Go” technique to introduce electronic components in additive manufacturing, rigorous quality control of the integrated electronic elements is necessary, as any failure can result in the complete loss of the part and an increase in manufacturing costs.

Based on the systematic review, it is concluded that although additive manufacturing is an intriguing process for manufacturing SSPs, there is still a need for significant advancement in critical areas such as material availability. It is essential for optimal performance of inductive sensors and SSPs themselves that a broader range of materials suitable for AM be developed.

In summary, additive manufacturing is a process that enables the fabrication of smart critical spare parts (SSPs), particularly suited for extreme operating conditions and real-time monitoring of production processes. However, there are areas for improvement and opportunities for future research that can further enhance and expand the use of additive manufacturing in the field of SSPs.

## 5. Conclusions

Additive manufacturing has emerged as a fundamental process in Industry 4.0, particularly over the last decade, enabling the production of functional parts with significant advancements. This method stands out for its flexibility and adaptability, particularly in the production of smart components. These parts not only perform specific functions but also gather and transmit key environmental data.

A specific example of SP application is the manufacturing of smart spare parts, essential in critical sectors such as energy [21], manufacturing [69] and aerospace [18,33]. These components have the capability to indicate the optimal time for their replacement, as well as predict and prevent catastrophic failures through continuous monitoring, which could represent a significant advancement. Although conventional manufacturing methods still dominate the market, additive manufacturing is well positioned to address specific niches, leveraging advantages such as customization and cost reduction.

It is important to note that according to the systematic review, this is a projection based on the identified potential. In the systematic search for smart parts, no clear studies or proposals were found. However, two relevant studies stand out outside of this search. The first study defines the concept of smart replacement parts under a logic of intelligent management systems, without highlighting design and manufacturing considerations or limitations [78]. Another study identifies manufacturing, design, and sensorization as fundamental aspects in the production of smart replacement parts, without specifying that these components must have the capability, through an integrated algorithm, to indicate the timing of their replacement and manage it accordingly [79].

However, significant challenges remain, such as optimizing the mechanical, thermal, electrical, and chemical properties of the materials used in this process. Several studies are focusing on the development of new materials for additive manufacturing, enhancing thermal treatments, and using advanced modeling tools to optimize not only mechanical behavior but also electrical and thermal properties. Overcoming these obstacles will be key to further expanding the application of additive manufacturing in the production of critical components.

Through a systematic review of scientific literature, it has been demonstrated that additive manufacturing provides a promising platform for the development of these technologies. The compilation and analysis of studies have provided substantial evidence regarding the benefits and challenges associated with implementing smart parts using this innovative technique.

## Figures and Tables

**Figure 1 sensors-24-05437-f001:**
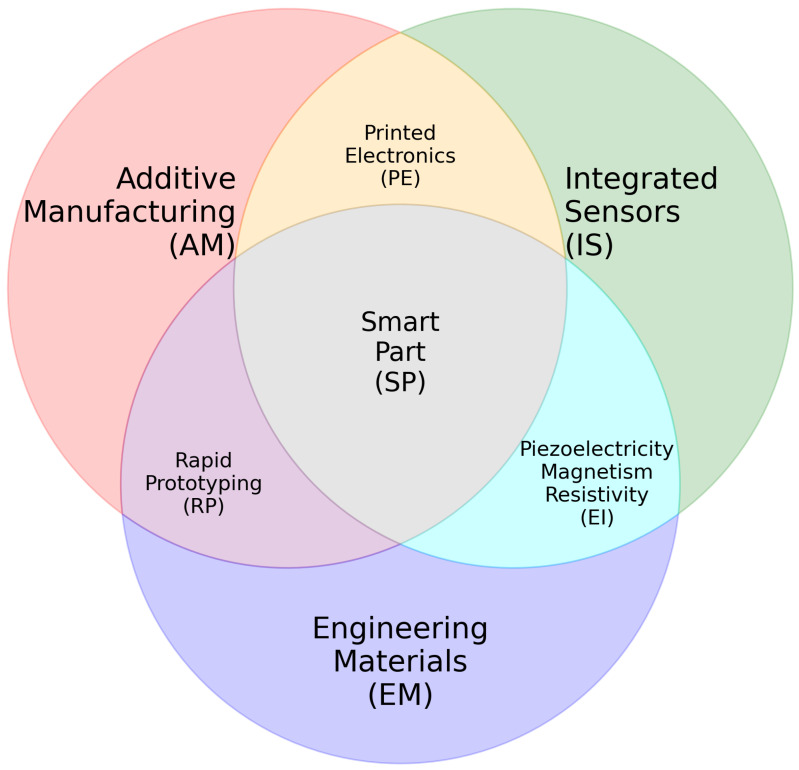
Venn diagram showing the convergence zone among different areas.

**Figure 2 sensors-24-05437-f002:**
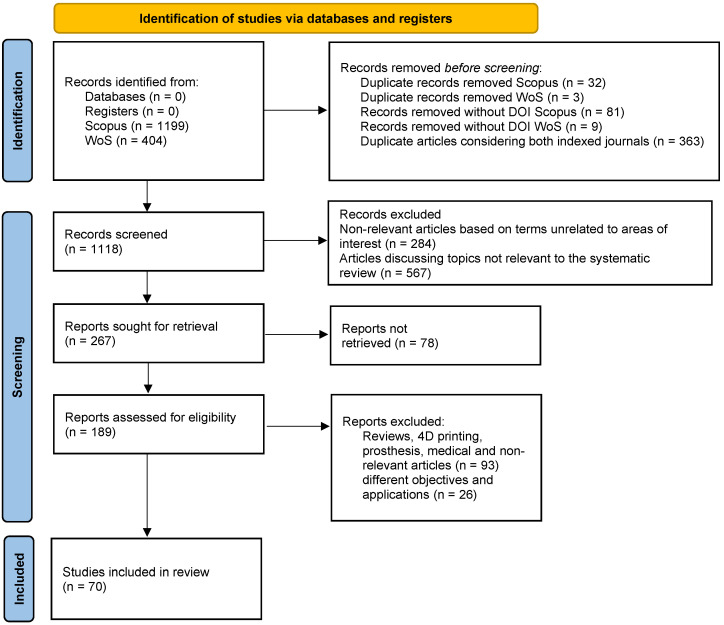
PRISMA flow diagram.

**Figure 3 sensors-24-05437-f003:**
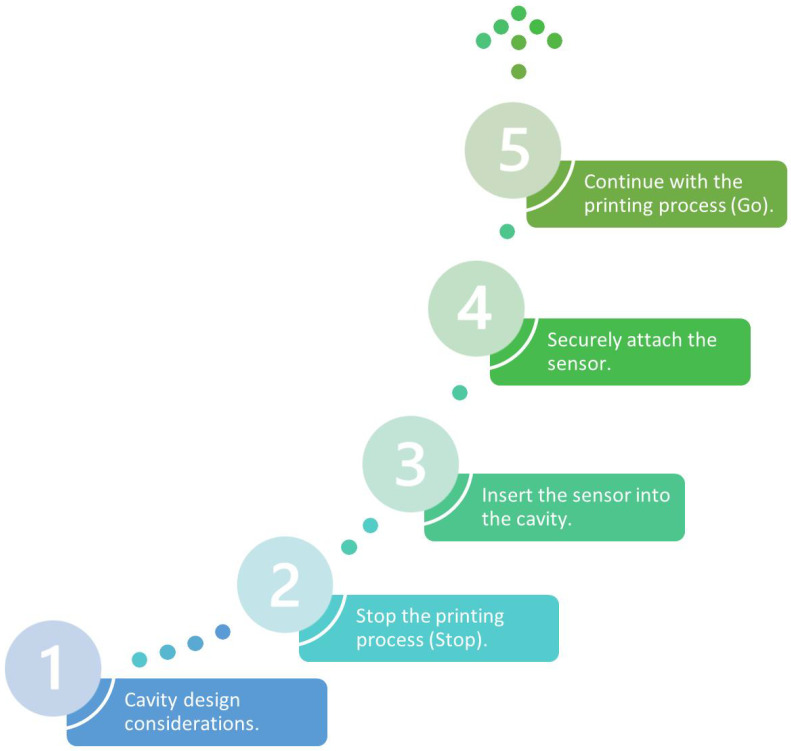
Stages for the introduction of a sensor inside the part.

**Figure 4 sensors-24-05437-f004:**
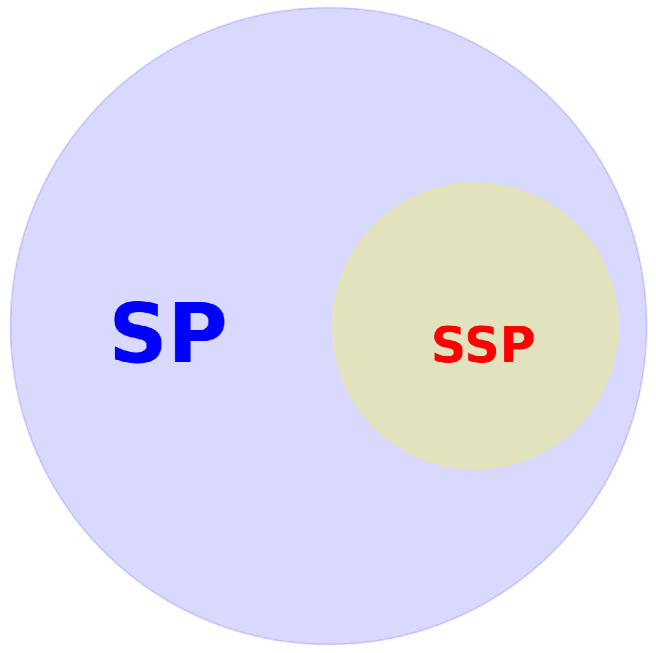
SSPs are considered a subcategory of SPs.

**Table 1 sensors-24-05437-t001:** Inclusion criteria for systematic review.

Criteria	Description
Time period	2010–2024
Language	English
Databases	Scopus and Web of Science
Document types	Journal articles, conference papers, reviews
Relationship with the research	Must align with the research questions, keywords, areas contributing to SPs

**Table 2 sensors-24-05437-t002:** Synonyms of Keywords in Specific Areas.

Specialty Area	Keyword Synonyms
Additive manufacturing (AM)	“Additive Manufacturing” or “3D Printing” or “Additive Fabrication”
Smart part (SP)	“Smart Part” or “Smart Spare Part” or “Intelligent Component”
Engineering materials (EM)	“Selection Material” or “Application Materials” or “3D Material”
Integrated sensor (IS)	“Sensor” or “Embedded Sensor” or “Incorporated Sensor” or “Included Sensor”

**Table 3 sensors-24-05437-t003:** Number of articles based on keyword combinations.

Keyword Combination by Area of Interest	No. of Articles in Scopus	No. of Articles in WoS
AM and SP	33	3
AM and EM	932	363
AM and IS	159	29
SP and EM	6	0
SP and IS	14	2
IS and EM	49	6
AM, SP, and IS	5	1
AM, SP, and EM	3	0
SP, EM, and IS	1	0
Total	1199	404

**Table 4 sensors-24-05437-t004:** Procedure for selecting articles to review.

Research Question	Symbol	Category of Analysis	Definition
RQ1 and RQ2	CA1	Manufacture of SP with AM	Discusses articles that have fabricated SPs and what are the considerations and limitations of fabricating SPs with AM.
RQ3 and RQ4	CA2	Incorporation of sensors and electronic components in AM for SP manufacturing.	The procedures and steps to introduce a sensor or an electronic component inside a part manufactured with AM are discussed.
RQ4 and RQ6	CA3	Scenario of SP manufacturing by AM and the impact on industrial processes.	The technical and economic impacts of including smart parts in production and industrial processes are discussed.
RQ5	CA4	Differences between an SP and a smart spare part (SSP)	The differences between an SP and an SSP are indicated

**Table 5 sensors-24-05437-t005:** Summary of AM considerations and limitations for SP manufacture.

Subcategory	Considerations	Limitations	References
Surface finish	There is flexibility in AM; parameters such as layer height and fill percentage can be reduced	Reduces mechanical, electrical, and electronic properties.	[8,9,13]
Material expansion and shrinkage	By considering the coefficient of thermal expansion in the design, damage can be avoided.	If there is no information on the coefficient of thermal expansion, tests must be carried out, increasing time and costs.	[7,15,20]
Mechanical performance	By increasing the percentage of filler and the layer height, higher mechanical performance can be obtained.	Post-treatment to increase mechanical properties may cause damage to the sensor.	[8,14,22,23,24]
Influence of the printing material on sensor performance	A variety of materials are available that do not reduce sensor performance.	Materials such as iron–carbon alloys can affect the performance of inductive sensors.	[8,25,26,27]
Influence of working and sintering temperatures on sensor performance	Sensors with a melting point higher than the sintering temperature should be processed.	Choosing sensors with melting temperatures close to those of the sensor can cause severe damage.	[7,28,29]
Materials with electrical and electronic properties	There are a variety of materials with electrical and electronic properties, but manufacturing parameters such as layer height and filler percentage must be considered for optimum performance.	Brittleness, thermal expansion, and melting point can lead to poor sensor or integrated-circuit performance.	[5,30,31,32,33]

**Table 6 sensors-24-05437-t006:** Summary of the difficulties of introducing sensors into SPs.

Steps	Description	Limitations	References
Cavity design considerations	The cavity must be larger and considering the expansion coefficients of the part and the sensor	There may be no information on the material, which can be solved by testing	[7,8]
Stopping the printing process (Stop)	The process must be stopped manually at the layer of interest; if it happens before it can cause the sensor to be above the level, it can cause damage to the part	The process is not automatic in many cases and there must be an operator performing the action manually	[8,21,34,35,36,37]
Insert the sensor into the cavity	The platform must be lowered using the printer commands. Once down, the sensor must be inserted very carefully	If the material is a polymer, ceramic, or polymer-based composite, it can generate distortions in the part, stresses, pickling, or warping	[8,21,34,35,36,37]
Influence of the printing material on sensor performance	A variety of materials are available that do not reduce sensor performance	Materials such as iron–carbon alloys can affect the performance of inductive sensors	[8,25,26,27]
Securely fasten the sensor	The sensor can be fixed with mechanical supports, soft solder, or glue	It should be noted that when using soft solder or glue, these may melt during sintering. Furthermore, these methods can only be applied to certain materials. As for the mechanical support, fasteners can be used, or the cavities can be designed in such a way that they help to securely fix the sensor	[8,21,34,35,36,37,38]
Continue with the printing process (Go)	Once the sensor is inserted, the operator must use the printer commands to restart printing	Avoid drafts or sudden temperature changes to reduce phenomena such as distortion, pickling, and warping	[8,21,34,35,36,37]

## Data Availability

No new data were created or analyzed in this study.

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
