# Peer review of "Smart Spare Parts (SSP) in the Context of Industry 4.0: A Systematic Review"

_sensors, 2024, doi:10.3390/s24165437_

Round 1

Reviewer 1 Report

Comments and Suggestions for Authors

The authors review the SP fabrication using additive manufacturing (AM), identifying the advantages and disadvantages of AM techniques in SP production and distinguishing between SP and smart spare parts (SSP). The review includes the methodology, the parameters of AM process and discusses the future work. This manuscript is qualified to be published.

Author Response

August 07th, 2024

Dear editor

Sensors

Re: Revised manuscript of paper: " E Smart Spare Parts (SSP) in the Context of Industry 4.0: A Systematic Review" by Morales Pavez G, Orlando Duran (your ref: (ID: Sensors-3108258).

I am writing to express my sincere gratitude for your thorough review and approval of my manuscript, [ID: Sensors-3108258]. I appreciate the time and effort you have dedicated to this process and am grateful for the opportunity.

Thank you once again for your approval and for facilitating the publication of my research.

Yours faithfully,

Gustavo Morales Pavez

Reviewer 2 Report

Comments and Suggestions for Authors
  • I found  strength of the papers
  • Comprehensive literature review with a systematic approach to selecting relevant articles.
  • Clear explanation of the advantages of additive manufacturing in producing smart parts and smart spare parts.
  • Emphasizing the potential of smart spare parts in critical industrial sectors and the benefits they offer in terms of predictive maintenance and cost reduction.
  • Recommendations for further research and development to address existing challenges in the field.
  • however main weakness is 
  • The paper could benefit from more specific examples or case studies demonstrating the practical application and impact of smart spare parts in different industries.
  • Further discussion on the limitations or drawbacks of additive manufacturing techniques in producing smart spare parts could provide a more balanced perspective.
  • i hav a few more questions which must be addressed carefully 
  • Could you provide more insights into the specific challenges faced in the development of smart spare parts (SSP) using additive manufacturing (AM)?

  •  
  • How do these challenges impact the overall feasibility and scalability of SSP production?

  • In your review, you mentioned the distinction between smart parts (SP) and smart spare parts (SSP). Could you elaborate on how SSP specifically replace conventional critical parts in industrial applications and offer enhanced functionality and reliability?

  • Can you provide examples of industries where SSP has shown significant benefits over traditional parts?

  • The systematic review resulted in 70 selected articles from an initial pool of 1,603.

  • Could you discuss the criteria used for excluding articles and how the final selection process ensured the relevance and quality of the chosen studies?

  • The conclusion highlights the potential of additive manufacturing in producing smart spare parts and its significance in critical sectors like energy, manufacturing, oil, and chemicals. Could you discuss any specific case studies or real-world applications where SSP has demonstrated a significant impact on efficiency, cost-effectiveness, or operational reliability?

  • The review mentions the challenges in optimizing mechanical, thermal, electrical, and chemical properties of materials used in additive manufacturing. Could you elaborate on how these challenges are being addressed in current research and development efforts, and what advancements can be expected in the future to overcome these obstacles?

  • best of luck 
Comments on the Quality of English Language

need major revisosns

Author Response

August 07th, 2024

Dear editor

Sensors

Re: Revised manuscript of paper: " E Smart Spare Parts (SSP) in the Context of Industry 4.0: A Systematic Review" by Morales Pavez G, Orlando Duran (your ref: (ID: Sensors-3108258).

Dear Editor

Hereby enclosed, and according to the comments by the reviewers. Jul 29th, 2024, we resubmit the manuscript that was submitted to Sensors (ID: Sensors-3108258). We gratefully acknowledge the valuable comments of your reviewers and have incorporated them, to the best of our intentions, into the revised manuscript in the following way:

Comments of Reviewer

We thank the reviewer for the careful reading of the manuscript.

  1. The paper could benefit from more specific examples or case studies demonstrating the practical application and impact of smart spare parts in different industries.

Answer: There has been discussion about the fabrication of spare parts using additive manufacturing, but there are no cases related to the production of smart spare parts. This study focuses on highlighting the various challenges that may arise in the fabrication of SSPs with additive manufacturing and detailing the characteristics that the component must have to implement the concept of a smart spare part.

There are studies that have ambiguously addressed the concept of smart replacement parts, associating it with an intelligent management system [80], but without detailing the technical aspects of manufacturing and the challenges involved. Other studies have addressed the concept in a general manner [81]. Additionally, the conclusions include two new studies that detail the development and application of smart parts, although without the ability to indicate replacement and merely monitoring the condition.

New text:

Chapter 5. Conclusion, pag. 19-17, line 635-643.

It is important to note that, according to the systematic review, this is a projection based on the identified potential. In the systematic search for smart parts, no clear studies or proposals were found. However, two relevant studies stand out outside of this search. The first study defines the concept of smart replacement parts under a logic of intelligent management systems, without highlighting design and manufacturing considerations or limitations [80]. Another study identifies manufacturing, design, and sensorization as fundamental aspects in the production of smart replacement parts, without specifying that these components must have the capability, through an integrated algorithm, to indicate the timing of their replacement and manage it accordingly [81].

  1. Further discussion on the limitations or drawbacks of additive manufacturing techniques in producing smart spare parts could provide a more balanced perspective.

Answer: the information was added, specified and clarified.

New text:

Chapter 4. Discussions and future work, pag. 19, line 595-611.

Furthermore, it is essential to discuss the limitations of additive manufacturing in the production of SSPs based on a systematic review:

Manufacturing Size: Additive manufacturing processes are limited by the size of the parts that can be produced. There are critical large-dimension components that cannot be manufactured through this process.

Materials: Although there is a wide range of materials that can be used in additive manufacturing, it is still insufficient for more critical applications, such as those requiring high temperature, strength, and pressure resistance.

Production Speed: Production speed is a limitation, as it is currently only suitable for the manufacturing of low-turnover parts.

Integration of Sensors and Electronic Components: The integration of sensors and electronic components within the parts may require multiple steps, increasing the likelihood of failure and thus extending manufacturing times.

Quality Control in the “Stop and Go” Technique: When applying the “Stop and Go” technique to introduce electronic components in additive manufacturing, rigorous quality control of the integrated electronic elements is necessary, as any failure can result in the complete loss of the part and an increase in manufacturing costs

  1. Could you provide more insights into the specific challenges faced in the development of smart spare parts (SSP) using additive manufacturing (AM)?

Answer: the information was added, specified and clarified.

New text:

Chapter 4. Discussions and future work, pag. 17-18, line 529-586.

  1. Material Limitations
  • Current Constraints: Additive manufacturing is an appealing process for the production of smart parts; however, it is currently constrained by the materials available in the market. The existing range of materials is often insufficient for high-temperature, high-strength, and high-pressure applications, which are critical for many SSPs.
  • Future Direction: Advancing new materials specifically designed for additive manufacturing will broaden manufacturing possibilities and enhance the performance of SSPs in demanding environments.
  1. Manufacturing Process Complexity
  • Current Issues: The manufacturing process for SSPs using AM involves multiple steps, including the integration of sensors and electronic components. Each step increases the potential for errors and extends production times.
  • Proposed Solutions: Reducing the number of steps in SSP manufacturing can increase production rates and minimize errors. Innovations such as the development of more efficient multimaterial 3D printers and the discovery of new materials can facilitate this step reduction and improve success rates in component manufacturing.

  • Sensor Integration Challenges
  • Design Constraints: The integration of sensors into SSPs without significantly altering the design poses a major challenge. Large or bulky sensors can compromise the structural integrity and functionality of the part.
  • Future Research: Research efforts should focus on designing smaller, more integrated sensors and electronic boards. This will enable the seamless incorporation of sensing capabilities into SSPs without compromising their design.
  1. Behavior in Extreme Conditions
  • Testing Limitations: While some studies have analyzed the behavior of SSPs in extreme conditions, such as high temperatures and pressures, there is a lack of comprehensive instrumentation protocols for such environments.
  • Research Needs: It is necessary to establish robust instrumentation protocols to ensure the reliable performance of SSPs under extreme conditions. This will involve extensive testing and validation of SSPs in real-world scenarios.

  1. Communication and Connectivity
  • Current Technology: Implementing the concept of smart spare parts (SSPs) using additive manufacturing requires the use of various predictive algorithms to determine the optimal replacement time. These algorithms are essential for real-time monitoring and lifecycle management of SSPs. Current challenges include the need for platforms capable of handling and analyzing large volumes of data generated by these predictive algorithms, ensuring precise and timely maintenance decisions.
  • Future Direction: Future innovations should focus on developing robust data analysis platforms that seamlessly integrate with predictive algorithms. These platforms must efficiently process vast datasets to enable accurate predictions for maintenance and replacement schedules. Additionally, designing smaller, more integrated sensors will enhance the data collection and transmission capabilities of SSPs, thereby improving their reliability and effectiveness within Industry 4.0 frameworks.
  1. Predictive Algorithms and Data Analysis
  • Current Technology: The communication of SSPs with equipment, instruments, controllers, and cloud-based systems (cloud computing) needs improvement. The integration of RFID antennas and other communication technologies is essential for the real-time monitoring and management of SSPs.
  • Material and Design Innovations: This requires the design and use of special materials that can operate under specific conditions, ensuring reliable communication and data transfer. Future innovations should focus on enhancing the connectivity and integration of SSPs within Industry 4.0 frameworks.

  1. How do these challenges impact the overall feasibility and scalability of SSP production?

Answer: the information was added, specified and clarified in the answer to question 3.

  1. In your review, you mentioned the distinction between smart parts (SP) and smart spare parts (SSP). Could you elaborate on how SSP specifically replace conventional critical parts in industrial applications and offer enhanced functionality and reliability?

Answer: the information was added, specified and clarified.

New text:

Subchapter 3.4. CA4: Differences between a Smart Part (SP) and a Smart Spare Part (SSP). Pag. 14-15, line 413-454.

The scientific articles analyzed in the systematic review define a sensorized part, which allows real-time monitoring of environmental conditions, as an SP [1], [69] Furthermore, by applying the same technique, it is possible to develop intelligent manufacturing processes, enhancing efficiency and real-time control of various variables, as demonstrated by the study on hybrid interconnections for 3D-printed electronics in harsh environments, as well as condition monitoring in foundry processes. Additionally, the data obtained by the sensors can be used to simulate the process, bringing it closer to reality [70], [71]. However, it is essential to discuss the definition of an intelligent part within a maintenance and condition monitoring environment, rather than in a general manner. This is because, considering the productive, economic, and safety impacts, the part must not only withstand environmental conditions but also have the capability to indicate the appropriate time for its replacement. Taking into account the aforementioned conditions and characteristics, we define an SSP.

These spare parts allow real-time monitoring of operating conditions, ensuring system integrity and performance in situations where conventional maintenance would be complicated or impossible [72]. Additionally, the mechanical component replaced by the SSP must be critical and interchangeable, as its failure in the production process would have significant economic and operational impacts. In summary, SSPs have the following characteristics:

  • They are sensorized and interchangeable parts.
  • They can transmit information wirelessly, reducing or eliminating wiring altogether.
  • They are implemented under extreme conditions such as high temperatures, high pressure, irradiation, among others.
  • They are manufactured using subtractive and additive manufacturing. The latter is attractive due to its manufacturing flexibility and low cost.
  • They allow real-time process monitoring.
  • They are highly critical parts, and their failure can significantly impact production processes due to their high costs.
  • They have the ability to indicate when they need to be replaced using algorithmic and predictive techniques, in conjunction with information obtained from sensors.
  • They reduce downtime and optimize costs by planning replacements, thereby allowing for adequate inventory management.

Based on the analysis conducted in the systematic review, it is concluded that all SSPs are SPs, but not all SPs are SSPs Figure 3. This implies that not all smart parts are designed to be spare parts. While some SPs may be integrated into a mechanism without the additional characteristics expected of an SSP, such as self-diagnostic capabilities to determine when they need to be replaced, others do possess these features. For instance, sensors can be integrated into a gear to monitor vibrations and send real-time data to prevent failures. In contrast, a gear that not only monitors its own condition and performance but also alerts and plans when it needs to be replaced, and can be managed within the inventory to ensure one is always available, exemplifies an SSP.

  1. Can you provide examples of industries where SSP has shown significant benefits over traditional parts?

Answer: I cannot answer this question with certainty, as it pertains to a concept defined in the study. Based on this definition, further work can be carried out focusing on the points mentioned in subsection 3.4 and chapter 4. These studies, which develop SP in the industry and are highlighted in the article ref: [1, 2, 7, 17, 21, 33, 71], can be emphasized.

  1. Could you discuss the criteria used for excluding articles and how the final selection process ensured the relevance and quality of the chosen studies?

Answer: the information was added.

New text:

Chapter 2. Methodology. Pag. 4-5, line 108-117

 In the identification phase, 398 duplicate articles were removed. Additionally, 90 articles without a DOI were discarded. In the second phase, the screening stage, 851 articles were eliminated based on their title and abstract. These articles did not contain the keywords for additive manufacturing and smart parts, reducing the total to 267 articles. In the "reports not retrieved" stage, 78 articles were removed due to access and payment restrictions, which was a critical phase as some of these articles were of interest. Finally, in the "reports assessed for eligibility" stage, 119 articles related to topics such as reviews, 4D printing, medicine, and other irrelevant subjects, as well as those with different objectives and applications, were eliminated. In the end, 70 articles were included in the study. The summary and the flow diagram are shown in Figure 2”.

Figure 2: Prisma Flow diagram

  1. The conclusion highlights the potential of additive manufacturing in producing smart spare parts and its significance in critical sectors like energy, manufacturing, oil, and chemicals. Could you discuss any specific case studies or real-world applications where SSP has demonstrated a significant impact on efficiency, cost-effectiveness, or operational reliability?

Answer:. Improvements are made in the conclusions, but specific examples cannot be added because the systematic review did not provide evidence for the proposed definition in the text. Consequently, the aim of this work is to encourage new research in this emerging field.

New text:

Chapter 5. Conclusion, pag. 19-20, line 628-643

A specific example of SP application is the manufacturing of smart spare parts, essential in critical sectors such as energy [21], manufacturing [71] and aerospace [17, 33]. These components have the capability to indicate the optimal time for their replacement, as well as predict and prevent catastrophic failures through continuous monitoring, which could represent a significant advancement. Although conventional manufacturing methods still dominate the market, additive manufacturing is well-positioned to address specific niches, leveraging advantages such as customization and cost reduction.

It is important to note that, according to the systematic review, this is a projection based on the identified potential. In the systematic search for smart parts, no clear studies or proposals were found. However, two relevant studies stand out outside of this search. The first study defines the concept of smart replacement parts under a logic of intelligent management systems, without highlighting design and manufacturing considerations or limitations [80]. Another study identifies manufacturing, design, and sensorization as fundamental aspects in the production of smart replacement parts, without specifying that these components must have the capability, through an integrated algorithm, to indicate the timing of their replacement and manage it accordingly [81].

  1. The review mentions the challenges in optimizing mechanical, thermal, electrical, and chemical properties of materials used in additive manufacturing. Could you elaborate on how these challenges are being addressed in current research and development efforts, and what advancements can be expected in the future to overcome these obstacles?

Answer: the information was added

New text:

Chapter 5. Conclusion, pag. 20, line 644-6450       

However, significant challenges remain, such as optimizing the mechanical, thermal, electrical, and chemical properties of the materials used in this process. Several studies are focusing on the development of new materials for additive manufacturing, enhancing thermal treatments, and using advanced modeling tools to optimize not only mechanical behavior but also electrical and thermal properties. Overcoming these obstacles will be key to further expanding the application of additive manufacturing in the production of critical components.

Yours faithfully,

Gustavo Morales Pavez

Reviewer 3 Report

Comments and Suggestions for Authors

The authors reviewed the literatures related to small parts, which is a relative new concept mainly related to additive manufactured parts with embedded electronics or sensors. The authors reviewed such research area in aspect of five points listed in Table 5.

Here are some comments:

1. The second paragraph in section 1 (line 32) is repeating contents mentioned in the first paragraph and can be removed.

2. Resolution of figures can be higher.

3. What is the meaning of PE, CA and "y" in Table 5? Please explain them so that readers can better understand.

4. In line 153, the authors define AM as a process "using a nozzle or a reloading arm". Such definition is not general enough since a lot of AM processes do not work like this, such as DLP, binder jetting, etc. Please define AM in a more inclusive way.

5. Section 3.1.2 mentioned some examples of material expansion and shrinkage for metal and ceramics. What about the literatures related to polymer materials and composites? Some polymer materials have relatively high melting temperatures and can also have such issues. The author can add those into this section.

6. In line 262, some liquid metals can also be electrically conductive for SP and beneficial for flexible circuits. Maybe the author can mention some literatures on that aspect.

7. For line 415, I checked the reference 70 and didn't see it mention the concept of small part (SP) there. Is it a concept created by the author, or it is truly from the literature? Please select the appropriate references if so.

8. After reading section 3.4, it is still not clear enough what is SSP. Please give more detailed examples (like a type of device or printed part) of SSP from research papers to explain this concept. It will also be helpful if the author can give some examples that are SP but not SSP to prove this concept. 

Comments on the Quality of English Language

The language of this manuscript needs further editing. Some places of the article did not use same language (Spanish mixed with English).

Author Response

August 07th, 2024

Dear editor

Sensors

Re: Revised manuscript of paper: " E Smart Spare Parts (SSP) in the Context of Industry 4.0: A Systematic Review" by Morales Pavez G, Orlando Duran (your ref: (ID: sensors-3108258).

Dear Editor

Hereby enclosed, and according to the comments by the reviewers. Jul 29th, 2024, we resubmit the manuscript that was submitted to Sensors (ID: Sensors-3108258). We gratefully acknowledge the valuable comments of your reviewers and have incorporated them, to the best of our intentions, into the revised manuscript in the following way:

Comments of Reviewer

We thank the reviewer for the careful reading of the manuscript.

  1. Another study that will be useful to discuss here The second paragraph in section 1 (line 32) is repeating contents mentioned in the first paragraph and can be removed.

Answer: Lines 19, 20, 21, and part of line 22 are removed. Additionally, lines 28 to 35 are deleted as they are repetitive.

New text:

Technological advances in areas such as additive manufacturing, electronics, and engineering materials allow for the implementation of new strategies in the production of smart parts (SP). An SP is characterized by its ability to perform a series of functions through the integration of sensors and communication systems. This enables it to provide data for wirelessly monitoring working conditions and anticipating catastrophic failures in production processes [1,2]. The design of a smart part (SP) must involve internal cavities that allow the integration of sensors and other electronic devices. However, the challenges posed by conventional manufacturing processes significantly limit the production of SPs, making it difficult to construct complex internal geometries and introduce sensors [1, 3, 4].

  1. Resolution of figures can be higuer.

Answer: The images were improved by increasing their size and resolution. Additionally, the acronym was changed from "EC" to "IS" to ensure consistency with the initials of each word

Figure 1. Venn Diagram Showing the Convergence Zone Among Different Areas.

Figure 2: Prisma flow diagram

Figure 3. Stages for the introduction of a sensor inside the part.

Figure 4. SSPs are considered a subcategory of SPs.

New word:

From “EC” to “IS”

  1. What is the meaning of PE, CA and "y" in Table 5? Please explain them so that readers can better understand

Answer: Changes are made to the acronyms to match the initials of each word. "SQ" (specific question) is replaced by "RQ" (research question) line 82-92. The acronym "CA" is specified in line 115. Additionally, the following tables are updated by adding the corresponding acronym: Table 2, Table 3, and Table 5.

New text:

Chapter 2. Methodology, pag. 3, line 82-92.

To limit the search, it is important to identify a series of research-related questions. This can be broken down into a general question (GQ) and research questions (RQ). Specifically:

  • GQ: What are the main challenges in implementing smart parts (SP) in Industry 4.0?
  • RQ1: What are the predominant factors that make additive manufacturing (AM) an ideal method for the construction of SP?
  • RQ2: What are the limitations of AM in the construction of SP?
  • RQ3: Will sensors and electronic components reduce their performance when incorpo rated into a part manufactured with AM?
  • RQ4: What types of materials are most suitable for the fabrication of SP using AM?
  • RQ5: Is there any substantial difference between SP and smart spare parts (SSP)?
  • RQ6: What economic impacts can SP generate in production processes?

New text:

Chapter 2. Methodology, pag. 5, line 118.

Based on the scientific articles obtained, the structure of the categories of analysis (CA) 115 that respond to the research questions is as follows.

  1. In line 153, the authors define AM as a process "using a nozzle or a reloading arm". Such definition is not general enough since a lot of AM processes do not work like this, such as DLP, binder jetting, etc. Please define AM in a more inclusive way

Answer: Information was added

New text:

Chapter 3. Results, pag. 6, line 147-154.

Additive manufacturing is a manufacturing process that involves the layer-by-layer deposition of material using various techniques, including but not limited to, methods that employ a nozzle or a reloading arm. The motions are typically performed in the x-y plane by a gantry containing a nozzle or laser. Once one of the layers is generated on the build platform, a movement driven by a servomotor allows the displacement in the z-axis of the platform, thus generating the successive layers that will form the 3D part. Other methods, such as DLP and binder jetting, also contribute to the diverse range of additive manufacturing techniques [10]

  1. Section 3.1.2 mentioned some examples of material expansion and shrinkage for metal and ceramics. What about the literatures related to polymer materials and composites? Some polymer materials have relatively high melting temperatures and can also have such issues. The author can add those into this section

Answer: The first paragraph addresses the general phenomenon. Study [14] analyzes the shrinkage of polypropylene (PP) and how to mitigate it to prevent parts from detaching. Study [20] examines the influence of volumetric expansion in the additive manufacturing process of sensors, using 1.75 mm PETG filament (MH Build Series, USA) to build the substrate and 1.75 mm Electrifi conductive filament (Multi 3D, USA) composed of polyester and copper to deposit traces. Finally, the next paragraph discusses a particular case in the manufacturing of a smart part and the design considerations needed to avoid damage to sensors and parts due to the different expansion coefficients of each material.

New text:

Subchapter 3.2. Material expansion and shrinkage , pag. 7, line 195-199.

All materials are sensitive to temperature changes, which can cause part distortion, cracking, dimensional losses, and metallurgical changes in the manufacturing processes. In additive manufacturing of smart parts, one notable occurrence is the expansion of polymeric and metal materials with increased temperature and their shrinkage when the temperature decreases.

  1. In line 262, some liquid metals can also be electrically conductive for SP and beneficial for flexible circuits. Maybe the author can mention some literatures on that aspect.

Answer: Information was added

New text:

Chapter 3. Results. Pag. 8, line 260-266.

Also, polymeric materials such as thermoplastic polyurethane (TPU), polydimethyl-siloxane (PDMS) and light-curing polymers are being used in flexible electronics. These materials enable the creation of electronic devices that are more versatile and adaptable to various applications, opening up new possibilities in the design and functionality of flexible technology. Additionally, a collaborative study by UC Berkeley and NCTU demonstrated the use of liquid metals to create passive components like resistors, inductors, and capacitors, highlighting their benefits for flexible circuits and SP applications [30].

  1. For line 415, I checked the reference 70 and didn't see it mention the concept of small part (SP) there. Is it a concept created by the author, or it is truly from the literature? Please select the appropriate references if so

Answer: The concept is a smart part. Corrections were made, and the paragraph was improved.

New text:

Subchapter 3.4. Differences between a Smart Part (SP) and Smart Spare Part (SSP). Pag. 14-15, line 155-161.

The scientific articles analyzed in the systematic review define a sensorized part, which allows real-time monitoring of environmental conditions, as an SP [1, 69]. Furthermore, by applying the same technique, it is possible to develop intelligent manufacturing processes, enhancing efficiency and real-time control of various variables, as demonstrated by the study on hybrid interconnections for 3D-printed electronics in harsh environments, as well as condition monitoring in foundry processes. Additionally, the data obtained by the sensors can be used to simulate the process, bringing it closer to reality [70, 71]. However, it is essential to discuss the definition of an intelligent part within a maintenance and condition monitoring environment, rather than in a general manner. This is because, considering the productive, economic, and safety impacts, the part must not only withstand environmental conditions but also have the capability to indicate the appropriate time for its replacement. Taking into account the aforementioned conditions and characteristics, we define an SSP.

  1. After reading section 3.4, it is still not clear enough what is SSP. Please give more detailed examples (like a type of device or printed part) of SSP from research papers to explain this concept. It will also be helpful if the author can give some examples that are SP but not SSP to prove this concept.

Answer: Description was added.

New text:Subchapter 3.4. CA4: Differences between a Smart Part (SP) and a Smart Spare Part (SSP). Pag.13-14, line 411-451.

The scientific articles analyzed in the systematic review define a sensorized part, which allows real-time monitoring of environmental conditions, as an SP [1], [69] Furthermore, by applying the same technique, it is possible to develop intelligent manufacturing processes, enhancing efficiency and real-time control of various variables, as demonstrated by the study on hybrid interconnections for 3D-printed electronics in harsh environments, as well as condition monitoring in foundry processes. Additionally, the data obtained by the sensors can be used to simulate the process, bringing it closer to reality [70], [71]. However, it is essential to discuss the definition of an intelligent part within a maintenance and condition monitoring environment, rather than in a general manner. This is because, considering the productive, economic, and safety impacts, the part must not only withstand environmental conditions but also have the capability to indicate the appropriate time for its replacement. Taking into account the aforementioned conditions and characteristics, we define an SSP.

These spare parts allow real-time monitoring of operating conditions, ensuring system integrity and performance in situations where conventional maintenance would be complicated or impossible [72]. Additionally, the mechanical component replaced by the SSP must be critical and interchangeable, as its failure in the production process would have significant economic and operational impacts. In summary, SSPs have the following characteristics:

  • They are sensorized and interchangeable parts.
  • They can transmit information wirelessly, reducing or eliminating wiring altogether.
  • They are implemented under extreme conditions such as high temperatures, high pressure, irradiation, among others.
  • They are manufactured using subtractive and additive manufacturing. The latter is attractive due to its manufacturing flexibility and low cost.
  • They allow real-time process monitoring.
  • They are highly critical parts, and their failure can significantly impact production processes due to their high costs.
  • They have the ability to indicate when they need to be replaced using algorithmic and predictive techniques, in conjunction with information obtained from sensors.
  • They reduce downtime and optimize costs by planning replacements, thereby allowing for adequate inventory management.

Based on the analysis conducted in the systematic review, it is concluded that all SSPs are SPs, but not all SPs are SSPs Figure 3. This implies that not all smart parts are designed to be spare parts. While some SPs may be integrated into a mechanism without the additional characteristics expected of an SSP, such as self-diagnostic capabilities to determine when they need to be replaced, others do possess these features. For instance, sensors can be integrated into a gear to monitor vibrations and send real-time data to prevent failures. In contrast, a gear that not only monitors its own condition and performance but also alerts and plans when it needs to be replaced, and can be managed within the inventory to ensure one is always available, exemplifies an SSP.

  1. The language of this manuscript needs further editing. Some places of the article did not use same language (Spanish mixed with English).

Answer: Comments and observations related to the translation and acronyms are appreciated. The document has been improved and corrected.

Yours faithfully,

Gustavo Morales Pavez

Round 2

Reviewer 2 Report

Comments and Suggestions for Authors

Good work. Now, the paper can be accepted. 

Comments on the Quality of English Language

minor 

Reviewer 3 Report

Comments and Suggestions for Authors

The newest revision looks good to me. I think it is ready for publication now.